# Intra- and Inter-Rater Reliability of Strength Measurements Using a Pull Hand-Held Dynamometer Fixed to the Examiner’s Body and Comparison with Push Dynamometry

**DOI:** 10.3390/diagnostics11071230

**Published:** 2021-07-08

**Authors:** Javier González-Rosalén, Josep Carles Benítez-Martínez, Francesc Medina-Mirapeix, Alba Cuerda-Del Pino, Antonio Cervelló, Rodrigo Martín-San Agustín

**Affiliations:** 1Department of Physiotherapy, University of Valencia, CP 46010 Valencia, Spain; javigonros96@gmail.com (J.G.-R.); cuerda@alumni.uv.es (A.C.-D.P.); rodrigo.martin@uv.es (R.M.-S.A.); 2Department of Physiotherapy, University of Murcia, CP 30100 Murcia, Spain; mirapeix@um.es; 3Department of Electronic Engineering, University of Valencia, CP 46100 Valencia, Spain; cervello.toni@gmail.com

**Keywords:** hand-held dynamometer, reliability, pull, push, lower limb, upper limb, strength

## Abstract

Hand held dynamometers (HHDs) are the most used method to measure strength in clinical sitting. There are two methods to realize the assessment: pull and push. The purpose of the present study was to evaluate the intra- and inter-rater reliability of a new measurement modality for pull HHD and to compare the inter-rater reliability and agreement of the measurements. Forty healthy subjects were evaluated by two assessors with different body composition and manual strength. Fifteen isometric tests were performed in two sessions with a one-week interval between them. Reliability was examined using the intra-class correlation (ICC) and the standard error of measurement (SEM). Agreement between raters was examined using paired *t*-tests. Intra- and inter-rater reliability for the tests performed with the pull HHD showed excellent values, with ICCs ranging from 0.991 to 0.998. For tests with values higher than 200 N, push HHD showed greater differences between raters than pull HHD. Pull HHD attached to the examiner’s body is a method with excellent reliability to measure isometric strength and showed better agreement between examiners, especially for those tests that showed high levels of strength. Pull HHD is a new alternative to perform isometric tests with less rater dependence.

## 1. Introduction

Quantifying the magnitude of strength is useful for rehabilitation programs, providing helpful information on setting target values, for setting up appropriate exercise loads, and the effectiveness and progress of treatment [1]. The evaluation of strength is one of the usual practices by health professionals to assess healthy individuals [2,3,4] and in the management patients with different lower limb or upper limb pathologies [5], such as knee osteoarthritis [6], rotator cuff injuries [7], and neurogenic thoracic outlet syndrome [8]. Among the tools to measure strength in clinical sitting, the most used is hand held dynamometers (HHDs) [9], since it has advantages such as portability, cost, and ease of use compared to other more expensive and less versatile methods (i.e., isokinetic dynamometer) [10]. In general, HHDs can be classified into two types, push or pull [1,11,12,13].

Push HDD consists of the patient having to push against the HHD, which is usually stabilized by the examiner’s hand and has been shown to be a reliable method [10,14]. This push mode has the disadvantage that examiner’s sex and strength influence the strength values (the reliability increases when the rater is stronger than the subject) [14]. Pull HHD consists of the patient pulling the HHD, which is generally attached to a rigid structure such as espalier, stretcher, or glass suction cup and showing to be also a reliable method [12,15,16,17,18]. Therefore, since this method requires a certain infrastructure and make it difficult to carry out certain movements, push HHDs is the type commonly used by clinicians.

To solve the requirement to a structure (e.g., wall bars or stretcher), we propose in this study a new measurement modality for pull HHD (fixing it to the examiner’s body). Our hypothesis is that by means of this fixation, the strength of the subject can be counteracted efficiently to have reliable strength values. If this is confirmed, we consider that it may be a more clinically feasible method for routine patient assessment than other pull HHD methods because it does not need infrastructures. Therefore, the main objective of this study was to evaluate the intra- and inter-rater reliability of this new measurement modality for pull HHD. Since previous studies have shown the influence of examiner’s strength on measurements with HHD, we used for our purpose two examiners and several muscles with different levels of strength. Furthermore, a secondary objective was to compare the inter-rater reliability and agreement of the measurements of this pull HHD method versus the push HHD method against the hand, as a common method of isometric strength measurement.

## 2. Materials and Methods

This cross-sectional study enrolled 40 healthy volunteer subjects who were recruited through advertising in Blasco Ibañez Campus of the University of Valencia (Table 1 shows the participants’ characteristics). The specific inclusion criteria were: (1) participants’ age between 18 and 40 years; (2) not having undergone a surgical operation on the lower or upper limb in the last two years; and (3) not having suffered pain episodes in the lower or upper limb two months before data collection. After a detailed explanation of the study procedures, the participants signed informed consent. The experimental protocol was approved by the Ethics Committee of the University of Valencia (Spain) (H1533739889520). Data collection was carried out in the clinical research laboratory of the Department of Physiotherapy (University of Valencia).

### 2.1. Procedures

Two sports and health professionals (a female and a male) were chosen to carry out the isometric tests, both with 1 year of clinical experience and with a master’s degree. Both raters, with different body composition (body mass: 55.4 kg and 91.3 kg and stature: 166 cm and 180 cm, respectively) were chosen to reflect different profiles of clinicians working both clinical and research settings. As previous authors [13], the raters completed a test of one maximum repetition of seated bench press as an indicator of general upper-extremity strength (47 kg for female tester and 81 kg for male tester). Raters received a 1-h training session on how to perform the measurements with both the pull HHD and the push HHD. Following the training, they performed testing procedures with 3 volunteers, supervised in turn by a health professional with extensive HHD experience. Both examiners were blinded to the strength values, with a third researcher responsible for viewing the strength values and recording them.

The pull HHD selected for the study was DiCI (Ionclinics S.L, L’Alcudia, Spain), which registers the traction strength through two hooks in series [19]. For the DiCI measurement, one end was attached with a strap to the subject’s ankle or wrist and the other end, with a belt, to the examiner’s body (Appendix A). On the other hand, the push HHD used was MicroFET2 (Hoggan Health Technologies Inc., Salt Lake City, UT, USA), widely used in the literature [20,21].

Isometric tests were performed on the dominant leg or arm in two sessions with a one-week interval between them. The two sessions began evaluating the strength using the pull HHD by tester 1 (male) and, thus, evaluating the intra-tester reliability (both intrasession and intersession). Subsequently, the isometric strength of the participants was again measured either by rater 1 or rater 2, randomly, with the pull HHD the first session and with the push HHD the second session in order to examine the inter-rater reliability of each HHD (Figure 1).

Before performing the isometric tests, the anthropometric characteristics of the participants were measured. A warm-up was performed on a bicycle with low resistance and at comfortable speed (80 revolutions per minute) for 10 min and three submaximal isometric contractions for each position. In addition, these submaximal contractions were also used to familiarize the participants with correct execution of the tests.

All tests were performed on a stretcher. The lower limb tests were performed both in the supine position for hip abduction (Hip-ABD), hip adduction (Hip-ADD), ankle flexion (Ank-F), and ankle extension (Ank-E) tests; in the prone position for hip extension (Hip-E), hip rotation external (Hip-ER), and internal (H-IR); and in the sitting position for hip flexion test (Hip-F). The upper limb tests were performed in supine for elbow flexion (Elb-F) and extension (Elb-E), for shoulder flexion (Sho-F), extension (Sho-E), and abduction (Sho-A), and for shoulder internal rotation (Sho-IR) and external (Sho-ER). These isometric tests (Figure A1 and Figure A2), both for lower and upper limb, were selected because they showed small measurement variation in previous studies [13,22,23]. Test order were randomized for each participant to avoid systematic bias related to this. Two 5 s MVICs were performed per movement with 60 s of rest between measurements. A rest of 10 min was applied between rater measurements. The participants were instructed to make the maximum effort and received oral motivations to maintain the strength performed.

### 2.2. Statistical Analysis

Participant characteristics and strength values (Newtons) are presented as mean ± standard deviation (SD) or percentages, as appropriate. The mean between repetitions was used for analyses. Custom written scripts computed with MATLAB (version R2019b; The Mathworks, Natick, MA, USA) was used to perform all statistical analyses by a researcher blinded for measurements.

First, for the analysis of the reliability, the values examined were: (1) the relative reliability using the intra-class correlation (ICC) and (2) the absolute reliability by calculating the standard error of measurement (SEM). The reliability was classified as excellent (ICC > 0.90), good (ICC = 0.76–0.90), moderate (ICC = 0.51–0.75), and poor (ICC < 0.50) [24,25]. SEM was calculated as SEM = $\mathrm{SD}\sqrt{\left(1-\mathrm{ICC}\right)}$.

Second, for the analysis of the agreement between the strength measurements (rater 1 and rater 2) and to assess systematic between-rater bias, that is, if values obtained by one rater systematically differed from that of the other rater, paired *t*-tests were used [26]. Furthermore, the differences between raters were calculated for each method and they were compared using paired *t*-test, with a level of significance *p* < 0.05. Additionally, to illustrate the differences between HHDs as a function of the strength obtained, the Bland Altmann plots were performed in those tests with higher strength values.

Sample size was calculated using the formula for reliability studies based on confidence intervals (CIs) described by [27]. With the number of instruments (k) equal to 2, the CI around r (the reliability coefficient) of 0.05, and an estimated r of 0.95, the sample size (*n*) was calculated to be 25 participants. However, ultimately, we included 15 more participants in the final sample in order to increase the study power.

## 3. Results

### 3.1. Intra-Rater Reliability

Table 2 shows the intra-rater reliability for the tests performed with the pull HHD, both intra-session and inter-sessions. The intra-session reliability showed excellent values, with ICCs ranging from 0.996 to 0.998. Furthermore, the SEM values were less than 1%. The inter-session reliability obtained similar values, with ICC higher than 0.995 and SEMs lower than 1%.

### 3.2. Inter-Rater Reliability

Table 3 shows the inter-rater reliability and agreement for the tests performed with the pull HHD. All tests showed excellent reliability (ICCs > 0.991), with SEMs lower than 1%. The agreement between rater showed differences between the measurements of rater 1 and rater 2 ranging from −0.69% to −3.78%, always in favor of rater 1.

Figure 2 illustrates differences between raters for the measurements of each participant, in the lower limb (Figure 2A) and the upper limb (Figure 2B). As can be seen, for some movements (e.g., hip abduction/adduction or hip rotations) both the pull and the push HHD methods showed differences lower than 20 N (rater differences ranged between 0.20% to 0.89% for pull HHD (Table 3) and between 0.26% to 1.59% for push HHD (Table 4)). On the other hand, for movements such as Hip-F, Ank-F, or Sho-E, both methods show greater differences between raters, but these are greater for the push HHD than for the pull HHD method; the differences between raters are −3.61%, −3.78%, and −2.84% for the pull HHD (Table 3) and −9.68%, −12.91%, and −9.71% for the push HHD (Table 4).

The differences by method as a function of the strength obtained in these three tests are illustrated in Figure 3 by means of the Bland Altman plots. Bland Altman plots show how from strength values greater than 200 N, the differences between raters for the push HHD increase progressively, while the differences in the pull HHD remain stable.

## 4. Discussion

Our results support our initial hypothesis that stabilizing a pull HHD to the examiner’s body has excellent reliability achieved for isometric strength measurements performed by examiners with different manual strength and in tests with different strength values. In addition, this new method presents a better agreement between examiners than push HHD against the hand, especially for tests with strength values greater than 200 N.

To our knowledge, this study is the first to examine the reliability of a pull HHD attached to the examiner’s body. Intra-reliability for this method proved to be excellent (ICCs > 0.998). Other studies with stabilized pull HHDs (these to a fixed external element) have also obtained high ICCs for intra-rater reliability, both for hip and ankle tests (ICCs ranged from 0.88 to 0.98) [12] and shoulder tests (ICCs ranged from 0.94 to 0.98) [16,17]. Otherwise, compared with other studies where they have used pull HHD attached to structures, our method showed ICCs for inter-rater reliability (ICCs > 0.991) similar or slightly superior to those studies (ICCs ranging from 0.69 to 0.99 for hip tests, from 0.76 to 0.99 for ankle tests, and from 0.86 to 98 for shoulder tests) [12,15,17,28]. Thus, the reliability of attaching a pull HHD to the examiner’s body would not be inferior to attaching it to a fixed external element. 

The agreement between the examiners’ measurements proved to be different between methods, especially in those tests with strength values greater than 200 N. As previous authors have described, in those tests with values greater than 200 N, the measurements of HHD without fixation compared to with fixation tend to underestimate the strength values [13,29]. Our results provide the novelty that fixing the HHD to the examiner’s body is sufficient to reduce such underestimation. For example, in tests such as Hip-F or Ank-F (with values close to or greater than 300 N), the differences between raters for the push HHD were 9.68% and 12.91%, respectively, compared to 3.61% and 3.78% for the pull HHD. In the upper limb this is similar, where the Sho-E, with values close to 300 N, showed differences between testers of 9.71% for push HHD versus 2.84% for pull HDD. The differences for push HHD between raters are similar to studies that have also used examiners with different strengths. For example, Kelln et al., 2008 found 8.87% differences for Ank-F [30]. 

This study proposes a new method to perform isometric tests, stabilizing an HHD pull in the examiner’s body. This method has shown excellent inter- and intra-rater reliability, and compared to other methods, its use provides clinical advantages for sports and health professionals. First, compared to other methods that have tried to solve the problem of examiner interaction (e.g., fixing the HHD to espalier, metal bar, or glass suction cup) [12,15,17,28,31,32], pull HHD method of this study presents a similar reliability to such methods but without subtracting clinical application as it does not need external fixation or is limited to specific movements. Second, pull HHD reduces the interaction of the examiner’s strength compared to the use of push HHD against the hand. Since push HHD is a common method of strength measurement among sports and health professionals due to its easy use, but it presents the bias of the examiner’s interaction, pull HHD fixed to the examiner’s body can be an alternative of easy use and less bias. Likewise, other types of clinical test has been used to assess the muscle performance, but with weak positive correlation against HHD [33]. 

This study had several strengths. First, we performed multiple tests of both lower limb and upper limb movements, eight and seven, respectively. To the best of our knowledge, this is the first study to examine the reliability of fifteen isometric tests, so we proposed a broad measurement protocol with HHD in the same study. Second, we avoided an information bias, since the raters were blinded from the strength values as there was a third researcher who was in charge of reading and recording them. In turn, a fourth researcher in charge of the statistical analysis was blinded as to which HHD corresponded to the different strength records.

The main limitation was that the measurements were made on a healthy population, limiting their generalization to other populations. Although it has been shown that the reliability of HHDs is lower in healthy population than in patients (due to greater strength and less variability), future studies should examine our protocol in clinical populations. Likewise, the inter-rater reliability was carried out by two raters, a procedure that according to the literature is sufficient for validation, but that the involvement of three or more raters might have provided even more reliable information. Future studies should address this limitation by considering, at least, three raters. Even so, we consider that this first study is essential to provide normative values in healthy people with which to compare.

## 5. Conclusions

This study examines the intra-and inter-rater reliability of a new proposal to measure isometric strength, a pull HHD attached to the examiner’s body. This method showed excellent reliability and acceptable agreement between the examiners’ measurements, who had a different body and strength profile. Furthermore, compared to the traditional method of strength measurement with HHD, pushing against the examiner’s hand, pull HHD showed better agreement between examiners, especially for those tests that showed high levels of strength. Thus, this new use of pull HHD may represent a new alternative for professionals who want to perform isometric tests with less influence of their strength on the values.

## Figures and Tables

**Figure 1 diagnostics-11-01230-f001:**
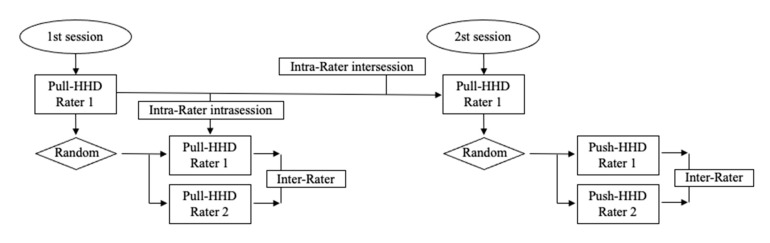
Flow chart represents a process of the study.

**Figure 2 diagnostics-11-01230-f002:**
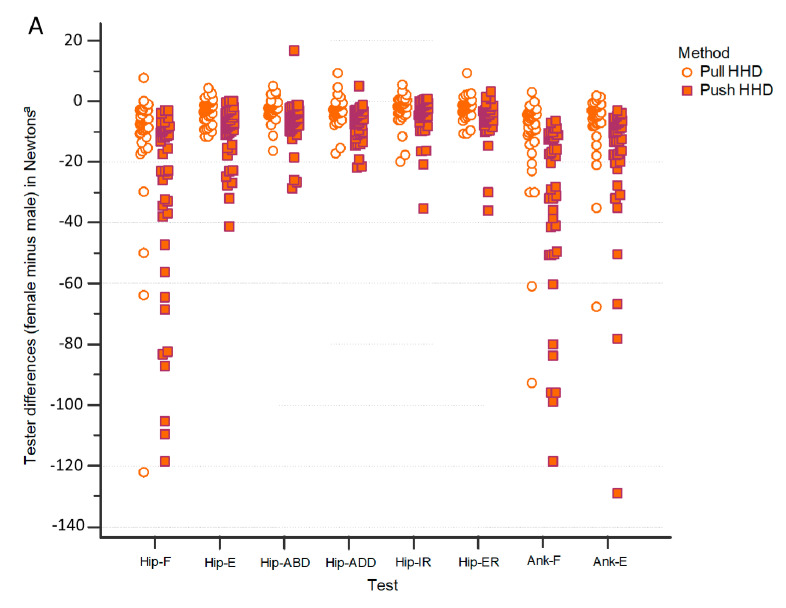
Differences between raters in the lower limb (**A**) and in the upper limb (**B**) for the strength measurements in each participant by test and method.

**Figure 3 diagnostics-11-01230-f003:**
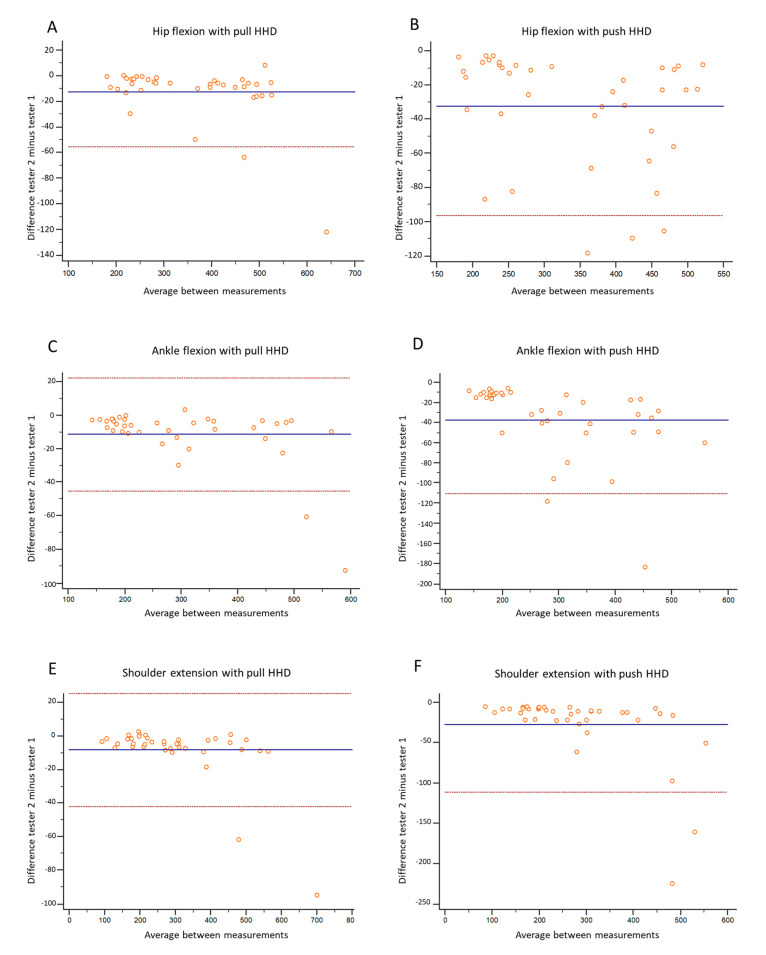
Bland Altman plots of the rater differences for pull HHD (**A**,**C**,**E**) and push HHD (**B**,**D**,**F**) methods as a function of total strength obtained in hip flexion (**A**,**B**), angle flexion (**C**,**D**), and shoulder extension (**E**,**F**).

**Table 1 diagnostics-11-01230-t001:** Characteristics of the participants (*n* = 40).

	Mean (SD)	Range
Age (years)	27.3 (5.1)	19–39
Body mass (kg)	66.2 (14.2)	47–94
Stature (cm)	170.7 (8.7)	152–192
BMI (kg/m^2^)	22.5 (3.4)	17.1–30.1
Gender	Males (*n* = 20)	Females (*n* = 20)

**Table 2 diagnostics-11-01230-t002:** Intra-rater reliability for the pull HHD method by movement and session.

		Intra-Session			Inter-Session		
Isometric Test		Test (SD)	ICC (95% CI)	SEM (%SEM)	Retest (SD)	ICC (95% CI)	SEM (%SEM)
Hip	Flexion	350.67 (125.19)	0.996 (0.993 to 0.999)	0.12 (0.03%)	350.49 (126.09)	0.998 (0.996 to 0.999)	0.14 (0.04%)
	Extension	210.32 (70.68)	0.997 (0.993 to 0.999)	0.09 (0.04%)	210.34 (71.5)	0.998 (0.996 to 0.999)	0.11 (0.05%)
	Abduction	129.72 (26.93)	0.998 (0.995 to 0.999)	0.05 (0.04%)	130.77 (27.61)	0.995 (0.992 to 0.998)	0.15 (0.12%)
	Adduction	144.88 (45.38)	0.998 (0.995 to 0.999)	0.07 (0.05%)	145.19 (45.77)	0.998 (0.995 to 0.999)	0.09 (0.06%)
	IN rotation	115.15 (33.87)	0.996 (0.992 to 0.999)	0.08 (0.07%)	115.11 (34.22)	0.996 (0.992 to 0.998)	0.20 (0.17%)
	EX rotation	129.34 (39.83)	0.998 (0.995 to 0.999)	0.06 (0.05%)	129.93 (39.32)	0.996 (0.993 to 0.998)	0.20 (0.16%)
Ankle	Flexion	305.85 (131.1)	0.998 (0.996 to 0.999)	0.10 (0.03%)	306.06 (131.81)	0.996 (0.993 to 0.999)	0.26 (0.08%)
	Extension	255.11 (116.06)	0.998 (0.996 to 0.999)	0.11 (0.04%)	256.04 (115.76)	0.998 (0.996 to 0.999)	0.10 (0.04%)
Shoulder	Flexion	212.91 (87.21)	0.997 (0.995 to 0.999)	0.13 (0.06%)	212.12 (87.02)	0.998 (0.995 to 0.999)	0.16 (0.07%)
	Extension	297.3 (141.38)	0.998 (0.996 to 0.999)	0.07 (0.02%)	298.41 (142.39)	0.997 (0.993 to 0.999)	0.14 (0.05%)
	Abduction	106.27 (40.21)	0.997 (0.994 to 0.999)	0.08 (0.07%)	106.52 (40.2)	0.998 (0.995 to 0.999)	0.08 (0.08%)
	IN rotation	141.32 (50.92)	0.998 (0.995 to 0.999)	0.09 (0.06%)	141.59 (51.39)	0.998 (0.995 to 0.999)	0.08 (0.06%)
	EX rotation	138.73 (45.26)	0.998 (0.995 to 0.999)	0.07 (0.05%)	138.99 (45.51)	0.996 (0.994 to 0.999)	0.14 (0.10%)
Elbow	Flexion	219.72 (113.09)	0.998 (0.996 to 0.999)	0.10 (0.04%)	219.13 (112.26)	0.998 (0.996 to 0.999)	0.17 (0.08%)
	Extension	161.26 (68.54)	0.998 (0.996 to 0.999)	0.08 (0.05%)	161.24 (68.3)	0.998 (0.995 to 0.999)	0.11 (0.07%)

SD = standard deviation; ICC = intra-class correlation; CI = confidence interval; SEM = standard error of measurement; IN = Internal; EX = External.

**Table 3 diagnostics-11-01230-t003:** Inter-rater reliability and agreement for pull HHD method by movement.

Isometric Test		Rater 1 (Male) Mean (SD)	Rater 2 (Female) Mean (SD)	ICC (95% CI)	SEM (%SEM)	Rater Differences (Rater 2 Minus Rater 1) Mean (%)	SD Difference Mean (%)
Hip	Flexion	351.92 (125.58)	339.22 (118.2)	0.992 (0.985 to 0.996)	1.95 (0.58%)	−12.70 (−3.61%)	21.82 (6.2%)
	Extension	210.28 (70.26)	208.84 (73.85)	0.991 (0.984 to 0.996)	1.26 (0.60%)	−1.44 (−0.69%)	13.24 (6.3%)
	Abduction	130.23 (26.94)	127.76 (26.16)	0.995 (0.991 to 0.998)	0.26 (0.20%)	−2.47 (−1.9%)	3.61 (2.77%)
	Adduction	145.33 (45.71)	142.34 (43.1)	0.998 (0.995 to 0.999)	0.19 (0.14%)	−2.99 (−2.06%)	4.36 (3%)
	IN rotation	115.68 (34.1)	113.19 (33.29)	0.995 (0.991 to 0.997)	0.33 (0.30%)	−2.49 (−2.15%)	4.73 (4.09%)
	EX rotation	129.62 (39.51)	127.53 (39.15)	0.998 (0.996 to 0.999)	1.13 (0.89%)	−2.08 (−1.61%)	3.57 (2.75%)
Ankle	Flexion	307.38 (130.95)	295.75 (123.9)	0.995 (0.991 to 0.998)	1.22 (0.41%)	−11.63 (−3.78%)	17.21 (5.6%)
	Extension	256.23 (116.67)	249.11 (113.42)	0.997 (0.995 to 0.999)	0.66 (0.26%)	−7.11 (−2.78%)	11.97 (4.67%)
Shoulder	Flexion	213.75 (87.17)	206.96 (83.27)	0.996 (0.993 to 0.998)	0.65 (0.31%)	−6.79 (−3.18%)	10.25 (4.79%)
	Extension	298.43 (141.48)	289.95 (132.62)	0.996 (0.993 to 0.998)	1.09 (0.37%)	−8.48 (−2.84%)	17.18 (5.76%)
	Abduction	106.83 (40.49)	104.96 (39.64)	0.999 (0.998 to 1)	0.07 (0.07%)	−1.87 (−1.75%)	2.31 (2.16%)
	IN rotation	139.37 (45.6)	139.03 (50.25)	0.999 (0.998 to 1)	0.09 (0.06%)	−2.79 (−1.97%)	2.86 (2.01%)
	EX rotation	141.82 (50.14)	136.53 (44.59)	0.997 (0.995 to 0.999)	0.25 (0.19%)	−2.84 (−2.04%)	4.63 (3.32%)
Elbow	Flexion	219.91 (112.98)	213.06 (106.79)	0.998 (0.996 to 0.999)	0.46 (0.22%)	−6.85 (−3.12%)	10.36 (4.71%)
	Extension	161.62 (68.68)	158.61 (67.36)	0.999 (0.998 to 1)	0.13 (0.08%)	−3.02 (−1.87%)	4 (2.47%)

SD = standard deviation; ICC = intra-class correlation; CI = confidence interval; SEM = standard error of measurement; IN = Internal; EX = External.

**Table 4 diagnostics-11-01230-t004:** Inter-rater reliability and agreement for the push HHD method by movement.

Isometric Test		Rater 1 (Male) Mean (SD)	Rater 2 (Female) Mean (SD)	ICC (95% CI)	SEM (%SEM)	Rater Differences (Rater 2 Minus Rater 1) Mean (%)	SD Difference Mean (%)
Hip	Flexion	337.38 (114.98)	304.74 (110.26)	0.979 (0.96 to 0.989)	4.72 (1.55%)	−32.64 (−9.68%)	32.57 (9.65%)
	Extension	203.01 (67.13)	195.45 (74.97)	0.966 (0.935 to 0.982)	4.77 (2.44%)	−7.56 (−3.72%)	25.87 (12.74%)
	Abduction	128.24 (25.14)	120.92 (24.19)	0.976 (0.955 to 0.987)	1.17 (0.97%)	−7.32 (−5.71%)	7.54 (5.88%)
	Adduction	142.61 (43.17)	134.92 (41.84)	0.996 (0.992 to 0.998)	0.35 (0.26%)	−7.69 (−5.39%)	5.55 (3.89%)
	IN rotation	115.80 (33.81)	110.55 (32.75)	0.989 (0.980 to 0.994)	0.72 (0.65%)	−5.24 (−4.53%)	6.84 (5.91%)
	EX rotation	128.79 (38.69)	122.69 (36.54)	0.991 (0.983 to 0.995)	1.95 (1.59%)	−6.10 (−4.74%)	7.15 (5.55%)
Ankle	Flexion	289.92 (114.83)	252.50 (103.33)	0.978 (0.943 to 0.984)	6.48 (2.57%)	−37.43 (−12.91%)	37.40 (12.90%)
	Extension	248.60 (106.19)	228.64 (103.26)	0.987 (0.975 to 0.993)	2.75 (1.20%)	−19.97 (−8.03%)	24.08 (9.69%)
Shoulder	Flexion	201.07 (79.36)	183.95 (70.6)	0.985 (0.971 to 0.992)	2.25 (1.22%)	−17.11 (−8.51%)	18.37 (9.14%)
	Extension	278.45 (124.56)	251.41 (107.06)	0.964 (0.933 to 0.981)	8.18 (3.25%)	−27.04 (−9.71%)	43.09 (15.47%)
	Abduction	103.24 (37.35)	97.82 (35.53)	0.996 (0.992 to 0.998)	0.30 (0.30%)	−5.42 (−5.25%)	4.72 (4.57%)
	IN rotation	139.51 (48.38)	132.03 (47.82)	0.991 (0.983 to 0.995)	0.43 (0.33%)	−7.48 (−5.36%)	4.49 (3.22%)
	EX rotation	136.70 (43.78)	130.50 (39.61)	0.998 (0.996 to 0.999)	0.35 (0.27%)	−6.20 (−4.53%)	7.92 (5.79%)
Elbow	Flexion	207.87 (99.62)	189.79 (89.26)	0.989 (0.979 to 0.994)	2.09 (1.10%)	−18.09 (−8.70%)	19.91 (9.58%)
	Extension	156.50 (65.6)	148.00 (64.08)	0.997 (0.995 to 0.998)	0.38 (0.26%)	−8.49 (−5.43%)	6.93 (4.43%)

SD = standard deviation; ICC = intra-class correlation; CI = confidence interval; SEM = standard error of measurement; IN = Internal; EX = External.

## Data Availability

The datasets generated during and/or analyzed during the current study are available from the corresponding authors on reasonable request.

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
