# Peer review of "Intra- and Inter-Rater Reliability of Strength Measurements Using a Pull Hand-Held Dynamometer Fixed to the Examiner’s Body and Comparison with Push Dynamometry"

_diagnostics, 2021, doi:10.3390/diagnostics11071230_

Round 1

Reviewer 1 Report

The study evaluates the intra- and inter-rater reliability of a new measurement modality for pull hand held dynamometer and to compare the inter-rater reliability and agreement of the measurements. The authors showed that pull hand held dynamometer attached to the examiner's body is a method with excellent reliability to measure isometric strength while provides better agreement between examiners, especially for those with high levels of strength.

Findings of this review are in my opinion of relevance to sport science and diagnostics and would fit the scope of the journal. There are, however, minor concerns which should be addressed.

I would suggest to reformulate the hypothesis. In the present form it seems that the reliability of measurement using the pull hand held dynamometer depends on the stability of the examiner.

Page 2, lines 49-50: Our hypothesis is that the examiner's body provides enough stability to have reliable strength values.

Please, include “Table 1. Characteristics of the participants (n=40)” in the section “2. Materials and Methods”

Rewrite the title of this section “3.1. Figures and Tables”

The authors should present the practical applications of obtained findings with respect to population tested.

The introduction is focused mainly on utilization of hand held dynamometers in clinical settings, whereas the author's recommendation is aimed at sports and health professionals.

Page 12, lines 264-266: Thus, this new use of pull HHD may represent a new alternative for sports and health professionals who want to perform isometric tests with less influence of their strength on the values.

Author Response

According to your recommendations, please find attached a copy of the revised version of our paper as a possible publication in DIAGNOSTICS.

The comments and suggestions for revision are reflected in the current version of the paper.  

  • Comment #1: The study evaluates the intra- and inter-rater reliability of a new measurement modality for pull hand held dynamometer and to compare the inter-rater reliability and agreement of the measurements. The authors showed that pull hand held dynamometer attached to the examiner's body is a method with excellent reliability to measure isometric strength while provides better agreement between examiners, especially for those with high levels of strength.

Findings of this review are in my opinion of relevance to sport science and diagnostics and would fit the scope of the journal. There are, however, minor concerns which should be addressed.

Response: We want to thank the reviewer 1 the consideration for the manuscript and its comments.

  • Comment #2: I would suggest to reformulate the hypothesis. In the present form it seems that the reliability of measurement using the pull hand held dynamometer depends on the stability of the examiner.

Page 2, lines 49-50: Our hypothesis is that the examiner's body provides enough stability to have reliable strength values.

Response: Thank you very much for your comment. Yes, we agree. We have reformulated the hypothesis to clarify it.

  • Comment #3: Please, include “Table 1. Characteristics of the participants (n=40)” in the section “2. Materials and Methods”

Response: Thank you very much for your comment. We have made the change.

  • Comment #4: Rewrite the title of this section “3.1. Figures and Tables”

Response: Thank you. We have rewritten the title.

  • Comment #5: The authors should present the practical applications of obtained findings with respect to population tested.

The introduction is focused mainly on utilization of hand held dynamometers in clinical settings, whereas the author's recommendation is aimed at sports and health professionals.

Page 12, lines 264-266: Thus, this new use of pull HHD may represent a new alternative for sports and health professionals who want to perform isometric tests with less influence of their strength on the values.

Response: Thank you very much for your comment. Yes, we agree with you. As you indicate, the introduction focuses mainly on clinical settings although the subjects of this study are healthy. Therefore, we in limitations statement "The main limitation was that the measurements were made on healthy population, limiting their generalization to other populations." We justify this below "Even so, we consider that this first study is essential to provide normative values in healthy people with which to compare."

Following your recommendations, we have modified the last sentence of clinical applications.

Reviewer 2 Report

Dear authors,

I recommend this article for publication. Current research problem is interesting for practice especially for sport. The paper has innovativity,, originality, rationality and completeness of research problem, originality of data, good tables and figures, good discussion and explanation of findings. It is clearly written. The methods are appropriate. The conclusions are reasonable. Writing style and language quality is at the academic level. Well done!

Author Response

According to your recommendations, please find attached a copy of the revised version of our paper as a possible publication in DIAGNOSTICS.

The comments and suggestions for revision are reflected in the current version of the paper.

REVIEWER 2

  • Comment #1: Dear authors,

I recommend this article for publication. Current research problem is interesting for practice especially for sport. The paper has innovativity, originality, rationality and completeness of research problem, originality of data, good tables and figures, good discussion and explanation of findings. It is clearly written. The methods are appropriate. The conclusions are reasonable. Writing style and language quality is at the academic level. Well done!

Response: We would like to thank Reviewer 2 for their consideration and all the comments made.

Reviewer 3 Report

The main limitation of the study was that the measurements were made by only two operators. This limits the generalization of the results and comprise the conclusion. The authors must increase the number of operators and of the subjects to have reliable results.
In this form the study cannot be considered for publication.

Author Response

  • Comment #1: The main limitation of the study was that the measurements were made by only two operators. This limits the generalization of the results and comprise the conclusion. The authors must increase (1a) the number of operators and (1b) of the subjects to have reliable results. In this form the study cannot be considered for publication.

Response: Thank you very much for your comments.

Comment (1a): We established two objectives for our study: (1) to evaluate the intra- and inter-rater reliability of this new measurement modality for pull HHD and (2) to compare the inter-rater reliability and agreement of the measurements of this pull HHD method versus the push HHD method against the hand. Intra- and inter-reliability of HHDs is a topic addressed by various authors in the literature, who generally used two examiners for this purpose:

Thorborg, K .; Bandholm, T .; Schick, M .; Jensen, J .; Hölmich, P. Hip Strength Assessment Using Handheld Dynamometry Is Subject to Intertester Bias When Testers Are of Different Sex and Strength. Scand J Med Sci Sports 2013, 23, 487–493, doi: 10.1111 / j.1600-0838.2011.01405.x.

Thorborg, K .; Bandholm, T .; Hölmich, P. Hip- and Knee-Strength Assessments Using a Hand-Held Dynamometer with External Belt-Fixation Are Inter-Tester Reliable. Knee Surg Sports Traumatol Arthrosc 2013, 21, 550–555, doi: 10.1007 / s00167-012-2115-2.

Alan McCall, Mathieu Nedelec, Christopher Carling, Franck Le Gall, Serge Berthoin & Gregory Dupont (2015): Reliability and sensitivity of a simple isometric posterior lower limb muscle test in professional football players, Journal of Sports Sciences.

Mentiplay, B.F .; Perraton, L.G .; Bower, K.J .; Adair, B .; Pua, Y.-H .; Williams, G.P.; McGaw, R .; Clark, R.A. Assessment of Lower Limb Muscle Strength and Power Using Hand-Held and Fixed Dynamometry: A Reliability and Validity Study. PLoS ONE 2015, 10, e0140822, doi: 10.1371 / journal.pone.0140822.

We think that considering this provided literature and the objective, two examiners are indicated for that purpose.

Comment (1b):  We have added the calculation of the sample size in statistical analysis.

Round 2

Reviewer 3 Report

Although the authors cited a few (old) articles that used only two physiotherapists to evaluate Inter- and intra-rater reliability of two  techniques, this is the minimum number that could be (maybe) appropriate in the sport science field, but not in diagnostic medicine. It should be increased to 4, atleast, to have robust data.
I appreciate that authors report the calculation of the sample size for the subjects, but it is not sufficient.

Author Response

Thank you very much for your help to improve our manuscript.

We have considered your recommendation and we have included in the limitations one sentence that reflects the problem about the raters’ number for reliability. On the other hand, in our study the question is how can influence the rater in the result get for the device and not the interpretation. Is for that, we considerer this study could help the health professionals in the clinical environment.

We are taking in count your recommendations for the next inter-rater reliability study.

We would like to show more (some of those found) updated studies with a similar topics who have used two raters to check the inter-rated reliability.

The authors

Elizagaray-García I, Gil-Martínez A, Navarro-Fernández G, et al. Inter, intra-examiner reliability and validity of inertial sensors to measure the active cervical range of motion in patients with primary headache. EXCLI J. 2021;20:879-893.

Ganderton C, Kerr B, King M, Lenssen R, Warby S, Munro D, Watson L, Balster S, Han J, Tirosh O. Intra-Rater and Inter-Rater Reliability of Hand-Held Dynamometry for Shoulder Strength Assessment in Circus Arts Students. Med Probl Perform Art. 2021 Jun;36(2):88-102.

McDaniel AT, Schroeder LH, Freedman JA, Wang Y, Heijnen MJH. Evaluating the Intra-Rater and Inter-Rater Reliability of Fixed Tension Scale Instrumentation for Determining Isometric Neck Strength. Int J Exerc Sci. 2021;14(3):563-577.